# Boosting PRRSV-Specific Cellular Immunity: The Immunological Profiling of an Fc-Fused Multi-CTL Epitope Vaccine in Mice

**DOI:** 10.3390/vetsci11060274

**Published:** 2024-06-15

**Authors:** Xinnuo Lei, Jinzhao Ban, Zhi Wu, Shinuo Cao, Mo Zhou, Li Zhang, Rui Zhu, Huipeng Lu, Shanyuan Zhu

**Affiliations:** 1Jiangsu Key Laboratory for High-Tech Research and Development of Veterinary Biopharmaceuticals, Engineering Technology Research Center for Modern Animal Science and Novel Veterinary Pharmaceutic Development, Jiangsu Agri-Animal Husbandry Vocational College, Taizhou 225300, China; xlei@jsahvc.edu.cn (X.L.); 2020107040@stu.njau.edu.cn (J.B.); zhiwu@jsahvc.edu.cn (Z.W.); shinuo_cao@jsahvc.edu.cn (S.C.); mo_zhou@jsahvc.edu.cn (M.Z.); lzhang@jsahvc.edu.cn (L.Z.); rzhu2021@jsahvc.edu.cn (R.Z.); luhuipeng@jsahvc.edu.cn (H.L.); 2Ministry of Agriculture Key Laboratory of Animal Bacteriology, International Joint Laboratory of Animal Health and Food Safety, College of Veterinary Medicine, Nanjing Agricultural University, Nanjing 210095, China

**Keywords:** porcine reproductive and respiratory syndrome virus (PRRSV), CTL epitope, Fc, cellular immunity, vaccine

## Abstract

**Simple Summary:**

This study introduces an innovative strategy for a cytotoxic T-lymphocyte (CTL) epitope-based multi-epitope vaccine to enhance cellular immunity against porcine reproductive and respiratory syndrome virus (PRRSV), aiming to surpass the limitations of existing vaccines. It pioneers the use of conserved CTL epitopes from PRRSV in vaccine development, eliciting a robust cellular immune response in a mouse model that makes it a potential vaccine candidate for the future.

**Abstract:**

The continuously evolving PRRSV has been plaguing pig farms worldwide for over 30 years, with conventional vaccines suffering from insufficient protection and biosecurity risks. To address these challenges, we identified 10 PRRSV-specific CTL epitopes through enzyme-linked immunospot assay (ELISPOT) and constructed a multi-epitope peptide (PTE) by linking them in tandem. This PTE was then fused with a modified porcine Fc molecule to create the recombinant protein pFc-PTE. Our findings indicate that pFc-PTE effectively stimulates PRRSV-infected specific splenic lymphocytes to secrete high levels of interferon-gamma (IFN-γ) and is predicted to be non-toxic and non-allergenic. Compared to PTE alone, pFc-PTE not only induced a comparable cellular immune response in mice but also extended the duration of the immune response to at least 10 weeks post-immunization. Additionally, pFc-PTE predominantly induced a Th1 immune response, suggesting its potential advantage in enhancing cellular immunity. Consequently, pFc-PTE holds promise as a novel, safe, and potent candidate vaccine for PRRSV and may also provide new perspectives for vaccine design against other viral diseases.

## 1. Introduction

Porcine reproductive and respiratory syndrome virus (PRRSV) is one of the most severe pathogens affecting the global pig industry. Since its initial report in the United States in the late 1980s, it continues to dominate the swine populations in most countries. PRRSV infection leads to reproductive and respiratory disorders, characterized by respiratory symptoms in piglets and reproductive failure, fetal death, and congenital infection in sows [1,2]. PRRSV is an enveloped, single-stranded, non-segmented, positive-sense RNA virus [3]. According to the latest 2022 classification by the International Committee on Taxonomy of Viruses (ICTV), PRRSV is now divided into two distinct species, namely *Betaarterivirus suid* 1 (formerly PRRSV-1, European genotype) and *Betaarterivirus suid* 2 (formerly PRRSV-2, North American genotype), both belonging to the *Betaarterivirus* genus (formerly *Porartevirus* genus), family *Arteriviridae*.

The current PRRSV vaccines on the market, which include inactivated and attenuated vaccines, are unable to provide sufficient protection against PRRSV infection. Inactivated vaccines present exogenous antigens, leading to antigen presentation primarily through the MHC class II pathway by antigen-presenting cells (APCs), which results in humoral immunity with a weak cellular immune response [4]. However, these vaccines elicit a comparatively weaker cellular immune response, which occurs through antigen cross-presentation via the MHC class I pathway. Attenuated PRRSV vaccines address these issues but raise a series of biosecurity concerns, such as the risk of vaccine-derived reversion to virulence, viral genome recombination, and virus shedding [5,6,7,8]. Moreover, both inactivated and attenuated PRRSV vaccines share the problem of inducing a large number of non-neutralizing antibodies, which can delay and reduce the production of neutralizing antibodies and may cause antibody-dependent enhancement (ADE) effects [8,9]. Therefore, there is an urgent need to develop more effective and safer PRRSV vaccines.

Epitope-based peptide vaccines, designed to target specific epitopes of pathogens, exhibit high specificity. B-cell epitope peptide vaccines can significantly enhance humoral immune responses but face challenges of insufficient cellular immunity and mutational escape [10]. In contrast, T-cell epitope peptide vaccines, especially cytotoxic T-lymphocyte (CTL) epitopes, elicit strong cellular immune responses that can cooperate with humoral immune responses, leading to the long-term success of vaccines [11]. Specifically, CTL epitopes promote the killing of virus-infected cells by CD8+ T cells through antigen cross-presentation, thereby enhancing the protective effect of the vaccine [12,13]. Furthermore, designing vaccines that target conserved epitopes is beneficial for developing a universal vaccine to overcome viral mutations.

Epitope-based peptide vaccines also possess additional advantages. They get rid of toxic components and decoy epitopes from the viral proteins, making them particularly suitable for combating viral infections that produce a large number of non-neutralizing antibodies and significant ADE effects [14]. Moreover, peptide vaccines are easy to design and produce, allowing for rapid adaptation to pathogen variation [15]. Additionally, they have shown great potential in the treatment of tumors and autoimmune diseases [16,17]. However, peptide vaccines have some drawbacks, such as poor targeting, short half-life, and low presentation efficiency [10]. To overcome these challenges, their performance can be enhanced by fusion with Fc molecules. The fusion of multi-epitope with the Fc molecules not only extends the plasma half-life by binding to Fc receptors for more sustained immunostimulation but also enhances cellular immunity by mediating antibody-dependent cellular cytotoxicity (ADCC) through FcγR or complement-dependent cytotoxicity (CDC) through complement C1q [18].

Our previous study (published in Chinese) constructed a fusion protein by fusing the neutralizing B-cell epitopes of PRRSV in tandem with the Fc molecule of mouse IgG. The fusion protein induced high-level humoral immunity and a number of neutralizing antibodies in mice and prolonged the presence of specific antibodies in the sera. To enhance the cellular immune response against PRRSV, in this study, we used enzyme-linked immunospot assay (ELISPOT) to screen multiple CTL epitopes of PRRSV, linked them in tandem, and then fused them with a modified porcine Fc molecule. Thereafter, the recombinant fusion protein was expressed in a prokaryotic system to achieve a high yield and high purity. Finally, the ability to induce cellular immune responses was evaluated by measuring cytokine levels. The aim of this study was to develop a novel anti-PRRSV epitope-based peptide vaccine to enhance cellular immune responses and overcome the limitations of existing vaccines. Our vaccine design strategy provides valuable insights for the development of next-generation PRRSV candidate vaccines and also serves as a reference for the design of vaccines against other viruses.

## 2. Materials and Methods

### 2.1. Virus Strains and Cell Lines

The PRRSV strain used in this study was isolated from the lung tissue of a pig with porcine reproductive and respiratory syndrome. The amino acid sequence of GP5 showed a 98.5% similarity to the GP5 sequence in the JXA1-R strain (vaccine strain), belonging to lineage 8.3, with three amino acid mutations at positions 35 (N to H), 59 (N to K), and 164 (R to G). MARC-145 cells were purchased from the BeNa Culture Collection (BNCC359828, Beijing, China) for PRRSV propagation and titration.

### 2.2. Animal Grouping and Immunization

Animal experiments were divided into two batches (Figure 1). The first batch of animals was used to evaluate the immunogenicity of synthetic peptides and recombinant proteins. Ten 4-week-old weaned piglets were purchased from an established PRRS-negative population (Large White/Duroc/Yorkshire hybrid breed). The pigs were confirmed to be double-negative for PRRSV nucleic acid and antibodies before immunization. They were divided into two groups: One group of five piglets was vaccinated with 1 × 10^4.5^ TCID50 of commercial attenuated vaccine (JXA1-R strain) via intramuscular injection. At three weeks post-vaccination, each piglet was challenged with 5 × 10^3.5^ TCID50 of the isolated JXA1 strain through intranasal (half-dose) and intramuscular (half-dose) routes. The other group was injected with PBS as a negative control. Peripheral venous blood was aseptically collected from the anterior vena cava three weeks post-challenge for the isolation of peripheral blood mononuclear cells (PBMCs). At six weeks post-challenge, the pigs were euthanized, and the spleen was harvested for the isolation of splenic lymphocytes. The second batch of animals was used to evaluate the immunological effects of the recombinant protein. Six-week-old SPF BALB/c female mice were divided into four groups of five, each vaccinated with PTE, pFc, and pFc-PTE, and the fourth group with PBS as a negative control. The immunization route was subcutaneous injection, with a recombinant protein dose of 20 μg/0.1 mL with Montanide™ Gel 01 ST adjuvant (SEPPIC), followed by a second immunization three weeks later with a halved dose. Blood and serum were collected from the mandibular vein of mice every 2 weeks for 10 weeks.

### 2.3. PBMCs and Splenic Lymphocyte Isolation

Briefly, 10 mL of anticoagulated blood from the anterior vena cava was diluted with 10 mL of D-Hanks buffer containing penicillin and streptomycin and then slowly layered onto 10 mL of lymphocyte separation medium and centrifuged at 800× *g* for 20 min. After centrifugation, the middle ring of the milky-white PBMC-enriched layer was aspirated, washed twice with an equal volume of D-Hanks buffer, and centrifuged at 250× *g* for 10 min to obtain PBMCs. The cells were resuspended in an RPMI 1640 medium, and cell viability was assessed using the trypan blue exclusion method followed by cell counting. After euthanizing the animals 7 weeks post-priming, the spleens were collected to assess the level of IFN-γ and IL-4 via ELISPOT. To isolate splenic lymphocytes, the spleens were aseptically removed, and a splenocyte suspension was prepared from the immunized PRRSV spleen in PBS. After the addition of 2 mL of red blood cell lysis buffer and centrifugation at 4 °C at 250× *g* for 10 min, the splenocytes were harvested. The splenocytes (1 × 10^6^ cells/mL) were plated in 24-well plates and cultured in an RPMI 1640 medium (Gibco, La Quinta, CA, USA) supplemented with 10% fetal bovine serum (Gibco) and 100 μg/mL penicillin–streptomycin (Hyclone, Logan, UT, USA).

### 2.4. ELISPOT and ELISA

The ELISPOT assay was conducted using Porcine IFN-γ and IL-4 ELISPOT kits (R&D Systems, Minneapolis, MN, USA) following the manufacturer’s protocols. Briefly, 1 × 10^6^ PBMCs were mixed with an equal volume of peptide (concentration 10 μg/mL) and added to a 96-well plate precoated with anti-IFN-γ antibody, with a volume of 100 μL per well. Positive controls included 10 μg/mL of phytohemagglutinin (PHA, Sigma-Aldrich, St. Louis, MO, USA) and PRRSV at a multiplicity of infection (MOI) of 1, while PBS served as a negative control. Cells were incubated at 37 °C with 5% CO_2_ for 24 h. After incubation, PBMCs were removed from the wells, and biotinylated detection antibodies, streptavidin-HRP, and TMB were used for subsequent incubations and color development. Once the PVDF membrane was air-dried, the spots were analyzed using an enzyme-linked spot analysis system. In the ELISPOT experiment with splenic lymphocytes, recombinant proteins PTE, pFc, and pFc-PTE were used for stimulation, and IFN-γ and IL-4 were measured, with other steps being essentially identical to those for PBMCs. The concentrations of cytokines in the sera were measured using mouse IL-2, IL-4, IL-5, and IL-10 ELISA kits (Solarbio, Beijing, China) and mouse IFN-γ and IL-12 ELISA kits (Beyotime, Shanghai, China) following the manufacturer’s protocols.

### 2.5. Peptide Screening and Synthesis

PRRSV encodes 16 non-structural proteins and 8 structural proteins. Based on the CTL epitopes collected from the literature, which are mainly distributed on 5 structural proteins (GP3, GP4, GP5, M, and N) and 2 non-structural proteins (Nsp2 and Nsp9) of different PRRSV genotypes, and considering the overlapping nature of some reported peptides as a single long peptide (10–15 amino acid residues), a total of 22 CTL epitopes were ultimately identified (Table 1). These epitope peptides were synthesized by GenScript (Nanjing, China) and confirmed to have a purity of over 95% by reverse-phase high-performance liquid chromatography (RP-HPLC) and mass spectrometry. Solubility tests were conducted in ddH_2_O, DPBS, or DMSO, with a storage concentration of 1–2 mg/mL.

### 2.6. Construction, Expression, and Purification of Fc Fusion Proteins

The recombinant plasmids pET30-PTE, pET30-pFc, and pET30-pFc-PTE were transformed into competent *Escherichia coli* BL21 (DE3) pLysS (Takara, Dalian, China). After identifying positive monoclonal colonies, the resulting expression strains were grown in Luria broth containing 50 μg/mL Kanamycin at 600 nm OD to 0.6–0.8 and then induced with 0.5 mM isopropyl β-D-1-thiogalactopyranoside at 37 °C for 3 h. Bacteria were collected by centrifugation and used for soluble protein analysis and purification. Sodium dodecyl sulfate–polyacrylamide gel electrophoresis (SDS-PAGE) was used to assess the expression levels of recombinant proteins in *E. coli*, followed by ultrasonic treatment in a lysis buffer (100 mM NaH_2_PO_4_·2H_2_O, 50 mM imidazole, 10 mM Tris-HCl, 300 mM NaCl, and 100 mM KCl, pH 8.5) with 0.5% Triton X-100 and 5 mM β-mercaptoethanol. The supernatant was then purified by Ni-NTA affinity chromatography (GE Healthcare, Buckinghamshire, UK). Finally, protein concentration was determined using the Pierce BCA protein assay kit (Thermo Fisher Scientific, Waltham, MA, USA).

### 2.7. Evaluation Software

The antigenicity of the recombinant proteins was predicted using VaxiJen v2.0 (http://www.ddg-pharmfac.net/vaxijen/VaxiJen/VaxiJen.html, accessed on 16 March 2023), with higher scores indicating greater antigenic potential. The immunogenicity of the recombinant proteins was predicted using the Immune Epitope Database (IEDB) Immunogenicity Server (http://tools.iedb.org/immunogenicity/, accessed on 18 March 2023), with higher scores indicating a greater likelihood of inducing an immune response. Further validation was performed using the ANTIGENpro server (http://scratch.proteomics. ics.uci.edu/, accessed on 18 March 2023). To better assess the potential allergenicity of the recombinant proteins, the AllerTOP v.2.0 server (http://www.ddg-pharmfac.net/AllerTOP/, accessed on 19 March 2023) was used, with predicted results categorized as “allergenic” or “non-allergenic.” Finally, the potential toxicity of the recombinant proteins was predicted using the ToxinPred2 server (https://webs.iiitd.edu.in/raghava/toxinpred2/batch.html, accessed on 19 March 2023), with results categorized as “non-toxic” or “toxic,” employing the Hybrid (RF + BLAST + MERCI) model with a threshold set at 0.6.

### 2.8. Statistical Analyses

Student’s *t*-test and one-way analysis of variance (ANOVA) with Tukey’s multiple-comparison test were applied to analyze the data in GraphPad Prism 6.0. The data are shown as means ± standard deviations (SDs). A *p*-value less than 0.05 was considered statistically significant (* *p* < 0.05, ** *p* < 0.01, and *** *p* < 0.001).

## 3. Results

### 3.1. The 10 CTL Epitopes Significantly Stimulated PBMCs to Release IFN-γ

A comparative analysis between the reported twenty-two CTL epitopes and the sequences of the PRRSV-2 epidemic strain JXA1 was performed, as detailed in Table 1. Thirteen corresponding sequences in the JXA1 strain were completely identical to the CTL epitopes, while the remaining nine sequences presented mutations at specific amino acid positions. To enhance the specificity of the peptides in the experiment, all the sequences used in this study were derived from the JXA1 strain. Subsequently, the online T-epitope prediction software NetMHCcons-1.1 was used to assess the binding affinity of the twenty-two CTL epitopes to the seven most abundant SLA alleles in hybrid pigs, as reported by Cao et al. [13] (Table 1). Eight CTL epitopes showed high binding affinity to specific SLA molecules, namely NSP9-TCE3, NSP3-TCE1, NSP3-TCE2, NSP4-TCE1, NSP4-TCE2, GP5-TCE1, M-TCE3, and N-TCE1. Another five epitopes showed weaker binding affinity, namely NSP2-TCE2, NSP9-TCE1, NSP9-TCE2, GP5-TCE3, and M-TCE2.

To evaluate the immunostimulatory activity of the twenty-two CTL epitopes, PBMCs were isolated from pigs three weeks after the PRRSV (JXA-1) challenge and then co-cultured with the individually synthesized peptides of CTL epitopes, followed by an ELISPOT assay to quantify INF-γ secreting cells. The results showed that ten CTL peptides could specifically stimulate the PBMCs, resulting in a considerable number of INF-γ secreting cells (>100 spots/1 × 10^6^) compared to the remaining epitopes (Figure 2A). This result was generally consistent with the predictions of NetMHCcons-1.1 (Table 1). Among them, NSP9-TCE3 and GP4-TCE2 were the most effective (>200 spots/1 × 10^6^). Notably, NSP9-TCE2 (YASAAAILM) was fully conserved across all the genotypes of PRRSV, which is crucial for the development of a universal vaccine against the highly mutable nature of PRRSV.

### 3.2. Design and Bioinformatics Analysis of Epitope Peptides Fused with Fc Protein

The design of the multi-epitope peptide was achieved by concatenating the 10 screened epitopes using either the flexible short linker GPGPG or AAY. Following evaluation with NetMHCcons-1.1, the concatenation strategy was designated as Poly-T-Epitopes (PTE) (Figure 2B). Subsequently, to construct the Fc fusion protein, the porcine IgG-derived (Uniprot: K7ZLA7) pFc molecule was retained with its CH2, CH3, and hinge regions. Moreover, two universal T-helper agonists, PADRE and TpD, were incorporated to enhance T-cell targeting efficiency [25]. Ultimately, pFc-PTE was generated by inserting PTE into pFc.

In addition, a bioinformatics analysis was conducted on the three recombinant proteins (Figure 2C), indicating a sequential decrease in antigenicity from PTE, pFc-PTE to pFc. Both PTE and pFc-PTE exhibited relatively high immunogenicity, while pFc demonstrated minimal immunogenicity, likely due to its nature as an endogenous protein in animals. Allergenicity prediction results showed that pFc-PTE and pFc are non-allergenic, whereas PTE was predicted to be allergenic. Toxicity prediction results indicated that all three proteins were non-toxic; however, PTE scored higher, indicating a greater potential for toxicity. Taken together, compared to the simple concatenation of epitope peptides, the fusion with pFc resulted in a reduction in the antigenicity (by 28%) and immunogenicity (by 51%) but improved the allergenicity and toxicity profiles of PTE (as evidenced by the numerical values).

### 3.3. The Recombinant Proteins Were Successfully Expressed and Purified

The encoding genes for PTE and pFc were codon-optimized and synthesized and then cloned into the pET30a vector. The construction of pET30-pFc-PTE was achieved through the enzymatic cleavage and ligation of PTE onto the pET30-pFc vector. Three recombinant plasmids were expressed in *E. coli* BL21(DE3) cells. The three proteins were then purified by Ni–NTA affinity chromatography (Figure 2D). Protein concentrations were determined using the BCA method, yielding peak concentrations of 1.28 mg/mL for PTE, 1.15 mg/mL for pFc, and 0.43 mg/mL for pFc-PTE. It is worth noting that the binding efficiency of pFc and pFc-PTE to nickel resin was not optimal, with much of the protein lost in the flow-through, suggesting that purification conditions could be optimized to improve protein yield.

### 3.4. pFc-PTE Stimulated PRRSV-Specific Splenic Lymphocytes to Secrete High Levels of IFN-γ

To determine whether the CTL epitope peptides retain the ability to be recognized by specific T cells after concatenation and fusion with pFc, the following experiment was conducted: The pigs immunized with the JXA1-R vaccine and later challenged with the JXA1 strain were euthanized at week 6. Splenic lymphocytes were harvested for in vitro stimulation with PTE, pFc, and pFc-PTE. Antigen specificity of T cells was measured by IFN-γ and IL-4 ELISPOT. Both PTE and pFc-PTE potently induced T cells to secrete IFN-γ and IL-4, with no significant difference in IFN-γ levels, while a significant difference was observed in IL-4 secretion (Figure 3A,B). These findings suggest that the fusion with pFc attenuates the Th2-inducing immunogenicity of PTE but does not compromise Th1 immune responses. Collectively, despite concatenation and pFc fusion, CTL epitope peptides remained recognizable by PRRSV-activated T cells, thereby retaining its immunogenicity.

### 3.5. pFc-PTE Induce a More Persistent Cellular Immune Response

To explore whether the fusion of pFc prolongs the clearance time of PTE in vivo, mice were immunized with PTE, pFc, and pFc-PTE. Subsequently, blood samples were collected, and the levels of IFN-γ and IL-4 in the sera were determined using ELISA kits. Within two weeks after the first immunization, the levels of IFN-γ and IL-4 in all experimental groups increased slightly, with no significant difference observed, suggesting the successful activation of the mice’s immune system by the recombinant proteins (Figure 3C,D). Notably, a pronounced increase in IFN-γ and IL-4 levels was detected at week 4 post-immunization, attributable to the booster immunization administered at week 3.

The PTE group exhibited peak cytokine levels at week 4 after initial immunization, while the pFc and pFc-PTE groups demonstrated peak levels at week 6. Subsequently, a sharp decline in IFN-γ and IL-4 levels was observed in the PTE group at weeks 6 and 8, contrasting with a more gradual decrease in the pFc and pFc-PTE groups, and it was not until at least 10 weeks after immunization that they returned to levels comparable to those in the PTE group. These findings imply that pFc fusion may enhance the persistence of PTE in vivo, providing a sustained stimulus for immune response induction. The prolonged response is likely due to the extended half-life of the pFc component, which facilitates enhanced antigen presentation and immune complex formation through its interaction with Fc receptors, thereby extending the immune response duration [26].

### 3.6. pFc-PTE Induced a Th1-Biased Immune Response

It is noteworthy that, in mice injected with PTE, pFc, and pFc-PTE, all three antigens elicited measurable levels of IFN-γ and IL-4 in the sera at 5 weeks after initial vaccination but with different peak values. The PTE group exhibited the highest peak of IFN-γ, with no significant difference from the pFc-PTE group, while the IFN-γ level in the pFc group was much lower than in the other two groups (Figure 4A). On the other hand, the PTE group also had a markedly higher peak of IL-4 than the pFc and pFc-PTE groups, with a minor and non-significant difference in IL-4 levels between the pFc and pFc-PTE groups (Figure 4B).

IFN-γ and IL-4 are cytokines characteristic of Th1 and Th2 cell types, respectively, with IFN-γ signifying cell-mediated immunity and IL-4 indicating humoral immunity [27]. To confirm that the immune response induced by pFc-PTE is predominantly of the Th1 type, we further measured the levels of four additional representative cytokines using ELISA. The changes in IL-2 and IL-12 (Th1-type cytokines) levels exhibited similar patterns to those of IFN-γ (Figure 4C,D). Additionally, the patterns of IL-5 and IL-10 (Th2-type cytokines) levels were consistent with those of IL-4 (Figure 4E,F). These findings indicate that pFc-PTE primarily elicits a Th1-biased immune response.

## 4. Discussion

PRRSV, as a major pathogen in the global pig industry, presents substantial challenges to vaccine research and development. The economic losses and concerns for animal welfare emphasize the critical need for novel vaccine development. Current vaccine strategies against PRRSV, including inactivated and attenuated vaccines, encounter several hurdles such as inadequate immunological protection, biosecurity risks, and the risk of ADE effects. These issues underscore the pressing need for innovative vaccine approaches. The present study introduced a pFc-PTE fusion protein designed to enhance and prolong the cellular immune response to PRRSV. This innovation significantly enhances both the theoretical and practical approaches to combating PRRSV in the swine industry.

Epitope vaccines, which offer better safety profiles than traditional live-attenuated vaccines, have been shown in numerous studies to have positive effects [28,29,30]. Additionally, epitope vaccines have a superior ability to cope with viral strain mutations compared to conventional vaccines [31]. This study addresses the limitations of epitope vaccines, including their suboptimal targeting and short half-life, by designing a fusion with the pFc. The Fc region targets immune cells expressing Fcγ receptors on their surface, thereby endowing the fused epitope peptides with robust immunogenicity and significantly enhancing the immune response of immune cells. Additionally, the production of these epitope peptides is rapid and straightforward, allowing for large-scale expression in *Escherichia coli*, which makes the production cost highly attractive.

Fc molecules have been applied in studies aimed at combating PRRSV infection. Studies have shown that the fusion of PRRSV receptors, sialoadhesin (Sn) and CD163, with Fc enhanced receptor half-life and cytotoxicity, conferring protection against PRRSV infection [32,33]. Additionally, Fc fusion has been demonstrated to enhance the immunogenicity of the PRRSV GP5 protein, eliciting the production of specific and neutralizing antibodies against PRRSV GP5 in mice [34]. Furthermore, Fc fusion facilitates the entry of PRRSV Nsp9-specific nanobodies into the monocyte–macrophage lineage cells, which highly express Fcγ receptors, thereby inhibiting PRRSV replication and extending the duration of action [35]. Building upon these studies, the present study introduces a novel approach by focusing on CTL epitopes, which are crucial for activating specific T-cell responses. The fusion with the Fc molecule not only enhanced the immunogenicity of PTE but also potentially improved its immunological effectiveness by prolonging the half-life and enhancing antigen presentation. Moreover, by using a series of bioinformatics analysis tools, we conducted a comprehensive assessment of the immunogenicity, antigenicity, and potential toxic side effects of the fusion protein, providing some predicted evidence for the safety of the vaccine.

pFc-PTE showed robust immunogenicity and sustained cellular immune responses in mice, and the results preliminarily demonstrated the feasibility of improving PRRSV cellular immune levels through the tandem fusion of CTL epitopes and Fc molecules. However, the translation of these findings to pigs is a critical next step. Our future studies will involve experiments in pigs to validate the immunogenicity and protective efficacy of pFc-PTE. That will provide valuable insights into the vaccine’s potential in a more clinically relevant setting. Additionally, establishing the optimal immunization dosage and delivery strategy, as well as evaluating long-term immunological effects and safety, requires further exploration. Furthermore, the cross-protective ability of the pFc-PTE fusion protein against different PRRSV variants has not been assessed, which is particularly important in the context of PRRSV’s high variability. There are currently seven representative strains from different genotypes of PRRSV (Lelystad virus, VR-2332, CH-1a, JXA1, NADC30, NADC34, and RFLP 1-4 lineage1C). Due to the high mutation rate of PRRSV, only NSP9-TCE2 (YASAAAILM) is fully conserved among these representative strains. Therefore, in order to improve the cross-protection ability against different PRRSV variants, it may be necessary to develop epitope pools that contain as many mutated epitopes as possible for immunization.

In summary, the pFc-PTE fusion protein shows potential in promoting Th1 cellular immune responses and prolonging the duration of the immune response, providing a new strategy for developing novel, safe, and highly effective PRRSV vaccines. Future research should focus on in vivo experiments, clinical evaluations, evaluating cross-protective capabilities, and assessing immunological effects against different PRRSV variants. Through these studies, a comprehensive understanding of the immunological and protective effects of the pFc-PTE fusion protein can be achieved, offering safer and more effective strategies for PRRSV control and inspiring new approaches to vaccine design for other viral diseases.

## Figures and Tables

**Figure 1 vetsci-11-00274-f001:**
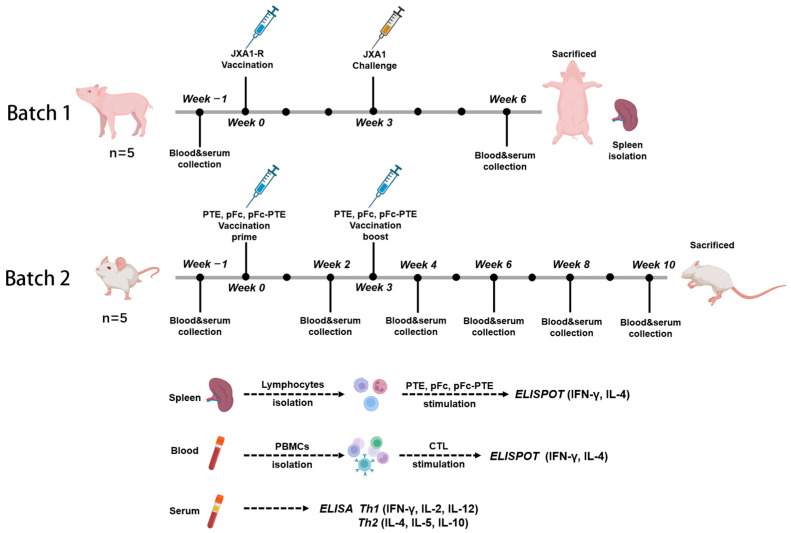
Schematic representation of animal grouping and immunization protocol.

**Figure 2 vetsci-11-00274-f002:**
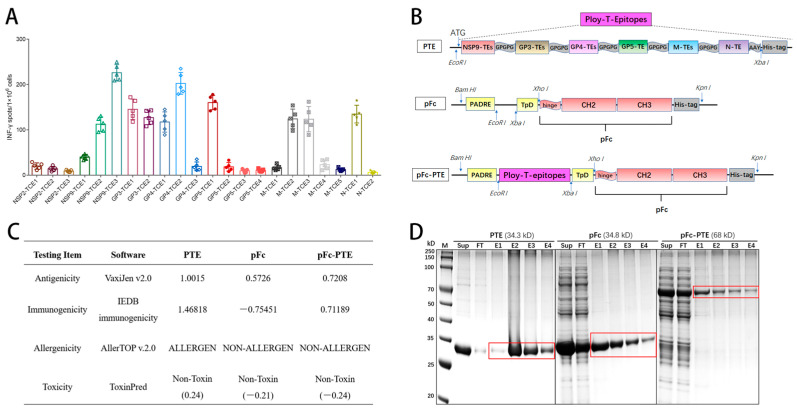
Construction, evaluation, and purification of recombination proteins: (**A**) Assessment of the immunostimulatory efficiency of the 22 CTL epitopes. After synthesis, these epitopes were used to stimulate PBMCs from experimental pigs (*n* = 5) that had been immunized with the JXA1-R vaccine and subsequently challenged with the JXA1 wild-type strain. The secretion of INF-γ was quantified by ELISPOT assay. (**B**) Schematic diagram of the genetic structures of the recombinant proteins. (**C**) Prediction of antigenicity, immunogenicity, allergenicity, and toxicity. (**D**) SDS-PAGE of purified proteins. Sup: supernatant after ultrasonic disruption and centrifugation. FT: flow-through (containing unbound proteins), E1–E4: eluted proteins, 1 mL/tube. The red wireframe indicates the location of the purified proteins.

**Figure 3 vetsci-11-00274-f003:**
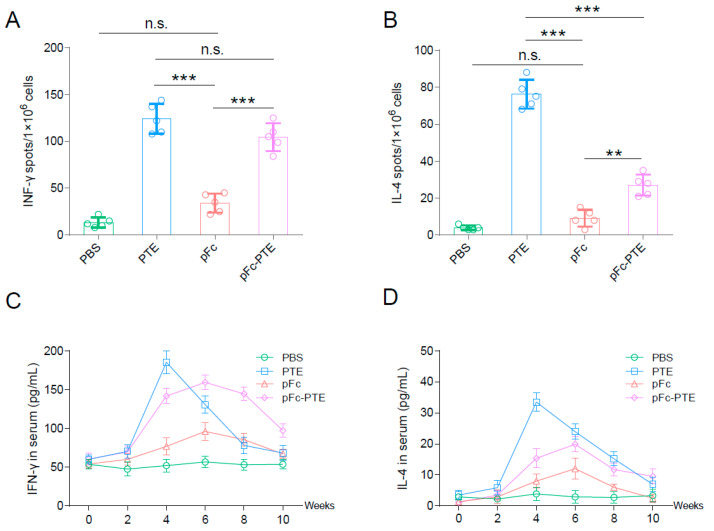
Recombinant proteins induce IFN-γ and IL-4 secretion from porcine splenocytes and mice: (**A**,**B**) ELISPOT analysis reveals IFN-γ and IL-4 secretion by PRRSV-specific splenocytes induced by PTE, pFc, and pFc-PTE. (**C**,**D**) Sustained cytokine production induced by recombinant proteins in mice. Bars show mean ± SD. ** *p* < 0.01, and *** *p* < 0.001, n.s. = not significant.

**Figure 4 vetsci-11-00274-f004:**
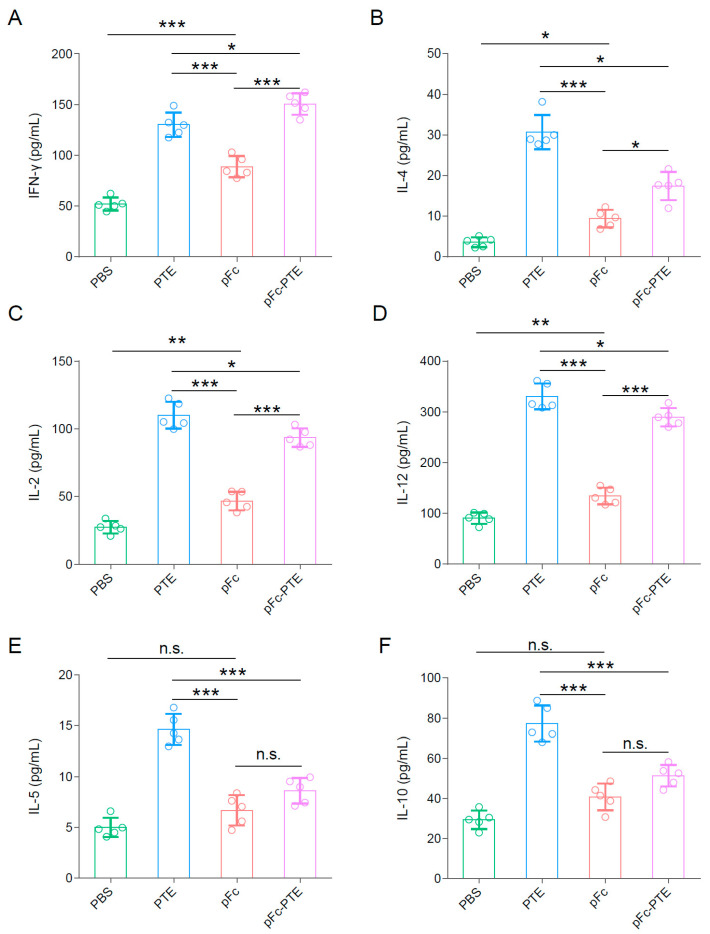
pFc-PTE induces a Th1-biased immune response: (**A**–**C**) Concentration of Th1 cytokines, IFN-γ, IL-2, and IL-12 induced by recombinant proteins in mice sera. (**D**–**F**) Concentration of Th2 cytokines, IL-4, IL-5, and IL-10 induced by recombinant proteins in mice sera. The tested sera were collected 21 days post-immunization. Bars show mean ± SD. * *p* < 0.05, ** *p* < 0.01, and *** *p* < 0.001, n.s. = not significant.

**Table 1 vetsci-11-00274-t001:** Summary and selection of PRRSV CTL epitopes.

CTL Epitope	Reported Sequence	Reference	Corresponding Sequence in JXA1	SLA-1 0401	SLA-1 0801	SLA-2 0101	SLA-2 0401	SLA-2 0502	SLA-2 1001	SLA-3 0401
NSP2-TCE1	QTLKLPAAL	[13]	QILRLPAAL	10	6	15	15	15	8	32
NSP2-TCE2	SIFQAPFTL		SAYQAFRIL	15	8	8	4	6	0.8	2
NSP2-TCE3	VVGPVGLGL		VVGPVGLGL	6	8	15	32	15	7	32
NSP9-TCE1	TMPPGFELY	[19]	TMPSGFELY	1	1.5	1.5	4	8	5	32
NSP9-TCE2	YASAAAILM	[13]	YASAAAILM	4	4	3	5	2	3	6
NSP9-TCE3	YSFPGPPFF		YSFPGPPFF	0.4	0.4	0.08	1	0.17	0.4	2
GP3-TCE1	ITAVYQTYY		ISAVFQTYY	0.8	0.8	0.4	0.08	5	5	3
GP3-TCE2	FSFELLVNY		FSFELMVNY	3	0.8	1.5	0.4	5	3	15
GP4-TCE1	MAASFLFLL		MAASFLFLL	9	8	2	7	0.08	5	4
GP4-TCE2	RTAIGTPVY		RTAIGTPVY	0.2	0.15	0.05	0.8	10	3	1.5
GP4-TCE3	CLFAILLAT	[20]	CLFAILLAI	32	9	32	32	6	10	32
GP5-TCE1	GLITVSAAGY	[13]	GLATVSTAGY	0.8	0.1	6	1.5	32	2	32
GP5-TCE2	LAALICFVIRLAKNC	[20,21]	LAALICFVIRLAKNC	50	50	50	50	32	50	50
GP5-TCE3	KGRLYRWRSPVIVEK	[21]	KGRLYRWRSPVIVEK	50	50	15	32	50	50	2
GP5-TCE4	FDWAVETFVLYPVAT	[22]	FDWAVETFVIFPVLT	50	50	50	50	50	50	50
M-TCE1	AVKQGVVNL	[23]	AVKQGVVNL	15	9	15	32	32	4	15
M-TCE2	YSAIETWKF		YSAIETWKF	1.5	3	1	4	0.8	6	15
M-TCE3	STNKVALTM	[13]	STNRVALTM	0.5	1.5	0.4	1.5	0.8	3	1
M-TCE4	KFITSRCRL	[24]	KFITSRCRL	32	32	10	50	32	32	5
M-TCE5	FGYMTFVHF		FGYMTFVHF	32	9	9	4	6	8	6
N-TCE1	FSLPTQHTV	[20]	FSLPTQHTV	15	15	9	16	0.3	15	9
N-TCE2	NPEKPHFPL		NPEKPHFPL	50	50	50	50	7	50	50

Note: In the corresponding sequence in JXA1, the mutated amino acids have been highlighted with a red font. NetMHCcons-1.1 (https://services.healthtech.dtu.dk/services/NetMHCcons-1.1/). Type of input: peptide; method = NetMHCpan; species = pig (SLA); allele = SLA-1*0401, SLA-1*0801, SLA-2*0101, SLA-2*0401, SLA-2*0502, SLA-2*1001, and SLA-3*0401. The threshold is based on %Rank; the smaller the value, the stronger the binding affinity, with ≤0.5 indicating strong binding affinity and 0.5~2 indicating weak binding affinity.

## Data Availability

The data presented in this study are available within the article.

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
