# Peer review of "Boosting PRRSV-Specific Cellular Immunity: The Immunological Profiling of an Fc-Fused Multi-CTL Epitope Vaccine in Mice"

_vetsci, 2024, doi:10.3390/vetsci11060274_

Round 1

Reviewer 1 Report

Comments and Suggestions for Authors

Dear authors, this is a quite interesting paper that proposes a novel vaccine design strategy. By fusing multi-epitope peptides with Fc, it aims to enhance cellular immune responses against PRRSV. The experimental design is logical and well-structured. The manuscript is well written, the figures are clear and well prepared. I strongly support that the manuscript should be published. However, I have a few concerns below:

Comment 1: In Table 1, does "consensus" refer to the reported CTL epitopes being conserved across various genotypes, or does it refer to the sequence consistency between the corresponding sequence in JXA1 and the reported CTL epitopes?

Comment 2: Line 261-263,the authors have concatenated the 10 identified CTL epitopes to construct PTE. In what order were these epitopes combined and why.

Comment 3: Theoretically, Fc molecules would form dimers through disulfide bonds, thereby stabilizing the structure of recombinant proteins. In this regard, eukaryotic systems seem more favorable for the expression of recombinant proteins. Why did the authors choose to use prokaryotic systems?

Comment 4: The main issue with current inactivated PRRSV vaccines is poor cellular immunity. Therefore, the concept of this study is excellent. It would be very meaningful if the immunological evaluation of this recombinant protein could be conducted through pigs rather than mice.

Reviewer 2 Report

Comments and Suggestions for Authors

The authors identified 10 PRRSV-specific CTL epitopes, constructed a multi-epitope peptide (PTE). And the cellular immune level of PTE vaccine was evaluated, and the effectiveness of the added Fc fragment was also evaluated. However, a major issue with this manuscript is the lack of evaluation of the effectiveness of animal testing for vaccine protection. In fact, these CTL epitopes have already been identified, and it is not significant to connect them in series for the determination of cellular immune related indicators. Moreover, there are numerous types of PRRSV strains in clinical, and the cross protection of the vaccine targeting JXA1 and its clinical value are questionable.

Comments on the Quality of English Language

Moderate editing of English language required

Reviewer 3 Report

Comments and Suggestions for Authors

Suggestions

May 30, 2024

Comments and Suggestions for Authors

Porcine reproductive and respiratory syndrome (PRRS) is an important disease that has cause huge economic losses to the pig industry. The current inactivated and attenuated PRRSV vaccines on the market cannot provide sufficient protection against PRRSV infection. This study introduces an innovative strategy for a cytotoxic T lymphocyte (CTL) epitope-based multi-epitope peptide vaccine to enhance cellular immunity against PRRSV. The pFc-PTE fusion protein as a candidate for PRRSV vaccines shows potential in promoting Th1 cellular immune responses and may provide a new strategy for developing novel, safe, and highly effective PRRSV vaccines. This is a good job.

Major concern

The robust immunogenicity and sustained cellular immune responses of pFc-PTE were obtained only from experiments in mice. It still needs experiments in pigs to validate the data. Please discuss this in the Part discussion.

Minor Points

 The writing could be improved. Some examples are as follows.

1.Line 84-88: Please re-write this sentence using short sentences.

2. Line 219:  Change “13 corresponding” to “Thirteen corresponding”.  

3. Line 225:  Change “8 CTL” to “eight CTL”.

Comments on the Quality of English Language

Comments and Suggestions for Authors

Porcine reproductive and respiratory syndrome (PRRS) is an important disease that has cause huge economic losses to the pig industry. The current inactivated and attenuated PRRSV vaccines on the market cannot provide sufficient protection against PRRSV infection. This study introduces an innovative strategy for a cytotoxic T lymphocyte (CTL) epitope-based multi-epitope peptide vaccine to enhance cellular immunity against PRRSV. The pFc-PTE fusion protein as a candidate for PRRSV vaccines shows potential in promoting Th1 cellular immune responses and may provide a new strategy for developing novel, safe, and highly effective PRRSV vaccines. This is a good job.

Major concern

The robust immunogenicity and sustained cellular immune responses of pFc-PTE were obtained only from experiments in mice. It still needs experiments in pigs to validate the data. Please discuss this in the Part discussion.

Minor Points

 The writing could be improved. Some examples are as follows.

1.Line 84-88: Please re-write this sentence using short sentences.

2. Line 219:  Change “13 corresponding” to “Thirteen corresponding”.  

3. Line 225:  Change “8 CTL” to “eight CTL”.

Reviewer 4 Report

Comments and Suggestions for Authors

The manuscript by Lei et al describes the use of tandem CTL epitopes with affinity to PRRSV JX-1 strain with the objective to explore the immunogenicity against PRRSV infection. The authors demonstrate the use of the Fc fusion construct to add longevity to the molecule but also to explore potency as a way of targeting additional immune cells upon inoculation. The work presented in this manuscript, offers valuable information in the realm of epitope-based vaccines. One of the most remarkable findings of this paper is the feasibility of including several CTL epitopes that are able to provide IFN -gamma responses and cytokines that are biased toward a Th-1 immunological responses from porcine PBMCs. However, the data and conclusion are in discordance. There are significant discrepancies between the graphs and what is written in the discussions and results. The authors need to address these issues. More importantly it is sad that the authors could not conduct these experiments in swine to determine the efficacy of these vaccine approaches.

Major

The title should be changed so that it really describes the immunogenicity profile done only in mice. The part done in pigs was to perform ex-vivo experiments

In the abstract section: “pFc-PTE not only induced a comparable cellular 24 immune response in mice but also extended the duration of the immune response…” the authors did not include for how long; this was missing throughout the manuscript.

The authors demonstrated that mice immunized with tandem CTL epitopes produced high levels of IFN-gamma from spleen-derived lymphocytes, this suggest that there is T cell activation, however a cytotoxic assay should have been performed to further evaluate this.

Were there any humoral responses? The authors should have conducted some PRRSV neutralizing antibody assays from the sera of inoculated mice.

In line 334: “Moreover, we conducted a comprehensive assessment of the immunogenicity, antigenicity, and potential toxic side effects of the fusion protein, providing additional assurance for the vaccine's safety” This is deceiving as they only conducted this via bioinformatics.

Minor

In line 339: “This prolonged response is likely due to the extended half-life of the pFc component, which facilitates enhanced antigen presentation and immune complex formation through its interaction with Fc receptors, thereby extending the immune response duration”. The authors should add a reference since they did not perform these experiments.

In line 401: “Furthermore, establishing the optimal immunization dosage and strategy, as well as evaluating long-term immunological effects and safety, require further exploration” What about route of inoculation? Would an oral inoculation improve the efficacy given that the route of infection starts in the upper respiratory system? If that is the case, wouldn’t IgA and mucosal immunity be more effective. The authors should discuss that.

Figure 3, the graphs are missing the X-axis labels. This becomes important for understanding the prolonged response.

Comments on the Quality of English Language

A new round of edits will be beneficial to this manuscript

Round 2

Reviewer 2 Report

Comments and Suggestions for Authors

The response is quite sufficient and there are no other comments.